# Bayesian Probabilistic Numerical Integration with Tree-Based Models

**Harrison Zhu, Xing Liu**
Imperial College London
{hbz15,xl6116}@ic.ac.uk

**Ruya Kang**
Brown University
ruya_kang@brown.edu

**Zhichao Shen**
University of Oxford
zhichao.shen@new.ox.ac.uk

**Seth Flaxman**
Imperial College London
s.flaxman@imperial.ac.uk

**François-Xavier Briol**
University College London
f.briol@ucl.ac.uk

## Abstract

Bayesian quadrature (BQ) is a method for solving numerical integration problems in a Bayesian manner, which allows users to quantify their uncertainty about the solution. The standard approach to BQ is based on a Gaussian process (GP) approximation of the integrand. As a result, BQ is inherently limited to cases where GP approximations can be done in an efficient manner, thus often prohibiting very high-dimensional or non-smooth target functions. This paper proposes to tackle this issue with a new Bayesian numerical integration algorithm based on Bayesian Additive Regression Trees (BART) priors, which we call BART-Int. BART priors are easy to tune and well-suited for discontinuous functions. We demonstrate that they also lend themselves naturally to a sequential design setting and that explicit convergence rates can be obtained in a variety of settings. The advantages and disadvantages of this new methodology are highlighted on a set of benchmark tests including the Genz functions, and on a Bayesian survey design problem.

## 1 Introduction

Numerical integration algorithms are key tools for modern statistics and machine learning. They allow us to tackle many of the integrals commonly arising in those fields and which cannot be computed in closed form. Examples include the computation of posterior expectations in Bayesian statistics, the marginalisation or conditioning of random variables, the computation of normalisation constants, calculations for the EM algorithm, or even the approximation of solutions to differential equations.

This paper will focus on the task of approximating the integral of some function $f : \mathcal{X} \to \mathbb{R}$ which is integrable with respect to some distribution $\Pi$ (whose density with respect to the Lebesgue measure is denoted $\pi$) over some domain $\mathcal{X} \subseteq \mathbb{R}^d$ ($d \in \mathbb{N}_+ = \mathbb{N} \backslash \{0\}$):

$$\Pi[f] := \int_{\mathcal{X}} f(x) d\Pi(x) = \int_{\mathcal{X}} f(x)\pi(x)dx. \qquad (1)$$

A large number of methods have been developed to tackle this problem. Classical quadrature rules [14] tend to be limited to low-dimensional problems and integrals against a small class of probability measures. Alternatively, Monte Carlo integration (MI) [68] only requires sampling independently from $\Pi$ and enjoys a convergence rate of $\mathcal{O}(n^{-1/2})$. When $\pi$ is unnormalised, sequential Monte Carlo (SMC) [15] or Markov chain Monte Carlo (MCMC) [68] samplers can be used and enjoy a similar rate. Quasi-Monte Carlo (QMC) [18] can improve on this convergence rate, but is limited to integration against a uniform measure in a hyper-cube (or simple transformations of this problem). We note that all of these methods are *quadrature rules*: they take the form $\hat{\Pi}[f] := \sum_{i=1}^n w_i f(x_i)$ where

$\{x_i\}_{i=1}^n \subset \mathcal{X}$ are design points and $\{w_i\}_{i=1}^n \subset \mathbb{R}$ are weights. Finally, the Laplace approximation [74] and variational inference [2] can be used, but these cannot be guaranteed to provide accurate estimates as $n$ grows.

Although all of the methods above are commonly used in practice, they all lack a straightforward non-asymptotic approach to quantifying our uncertainty about the value of $\Pi[f]$ after a finite number of function evaluations. This is not necessarily a problem for applications where $n$ can be taken to be large relative to the difficulty of the problem. However, those methods will be sub-optimal when $f$ is both difficult to approximate and expensive to evaluate since the asymptotic results will not hold.

An alternative approach comes from the field of probabilistic numerics [17; 35; 46; 58; 77], and in particular Bayesian probabilistic numerical methods [12], which frame problems in numerical analysis as statistical estimation tasks. This allows for the quantification of uncertainty surrounding the value of a quantity of interest using probabilistic statements valid for finite $n$. For integration, the main approach is called Bayesian quadrature (BQ) [4; 17; 57; 67]. Its name comes from the fact that the estimator takes the form of a quadrature rule, but also carries a Bayesian interpretation. However, we will see that not all Bayesian estimators have this property. For this reason, we prefer the general name of *Bayesian probabilistic numerical integration (BPNI)*.

BPNI starts by positing a prior distribution for the integrand $f$. It then computes the posterior distribution given values of $f$ at some design points, and finally considers the implied distribution on $\Pi[f]$. The main advantage of BPNI is that all our knowledge about the problem can be straightforwardly implemented in a prior, and the posterior distribution on $\Pi[f]$ can allow us to quantify our uncertainty about the exact value of this quantity. BPNI critically relies upon a flexible statistical model for the integrand $f$, which is almost always chosen to be a Gaussian process (GP). In this case, we have a quadrature rule which we will call GP-BQ. GPs have the convenient property that the posterior distribution can be obtained in closed form for interpolation and regression with Gaussian noise. They also have well-studied concentration rates which carry over to the corresponding GP-BQ methods [4; 41; 42; 82].

GP-BQ has received a lot of attention in recent years. Efficient deterministic [43; 44], adaptive [3; 23; 31; 40] and randomised [1] point-selection schemes have been designed. The method was extended to cases where $\pi$ is not available [55], and when multiple integrals are computed simultaneously [29; 83]. It has been applied to fields ranging from econometrics [56] to computer graphics [5] and robotics [65]. While GPs are virtually the only choice that has been explored in the BPNI literature, in practice they suffer from a number of challenges. These include:

1. **Discontinuities:** GPs are not suited for integration problems where the integrand is discontinuous – a common challenge in applications [13; 52]. As a result, they may lead to poor predictions and unreliable uncertainty estimates for these problems.

2. **Computational cost:** A major issue with GPs is their cubic cost in the number of data points. This can be mitigated using approximate GPs, but these do not necessarily lead to closed-form estimators. Exceptions include the work of [43] and [38] which allow for near-linear computational cost, but require $\Pi$ and $\{x_i\}_{i=1}^n$ to exhibit symmetry properties which may not hold in practice.

3. **High dimensions:** Few applications of GP-BQ exists where $d > 10$ due to the curse of dimensionality. This issue is closely linked to the computational cost issue since the number of points needed to approximate functions well will grow exponentially with dimension. However, it is sometimes possible to use sparsity of $f$ to lower this cost [4].

To tackle some of these issues, we propose to use an alternative model based on trees. The specific model is called Bayesian additive regression trees (BART) [10; 36], and as a result we call this BPNI method *BART integration (BART-Int)*. BART is a sum-of-trees model similar in spirit and effectiveness to random forests. It has been successfully deployed in a number of settings including causal inference [21; 37], genomics [19], behavioural sciences [84] and Bayesian optimisation [11]. Extensive experimental results in the literature have shown that BART does not usually overfit, thanks to its ensemble structure, and that it only requires a limited amount of hyperparameter tuning.

***Contributions of the paper:*** This paper derives a novel BPNI algorithm based on Bayesian priors with a tree structure. We show that BART is a natural choice for BPNI which can be preferable to GPs in any of the three settings highlighted. This is done through a mix of theory and experiments. On the theoretical side, we prove asymptotic convergence rates for BPNI with MCMC approximations. This

result may be of independent interest since it is applicable for any nonparametric method for which a concentration rate is known. On the experimental side, we compare our method to GP-BQ on a range of problems including the Genz test functions, a rare-event simulation problem and a Bayesian survey design problem.

## 2  Background: Bayesian Regression with GPs and BART

Before presenting BART-Int, this section introduces background material on Bayesian regression with GPs and BART. Given data $(X, y)$ consisting of design points $X = (x_1, \ldots, x_n)^\top \subset \mathcal{X}^n$ and responses $y = (y_1, \ldots, y_n)^\top \subset \mathbb{R}^n$, the regression problem is to recover an unknown function $f : \mathcal{X} \to \mathbb{R}$ for which we have access to noisy measurements $y$. These are usually assumed to be independently and identically distributed (i.i.d.) Gaussian: $y_i = f(x_i) + \epsilon_i$, $\epsilon_i \sim \mathcal{N}(0, \sigma^2)$ for $i = 1, \ldots, n$, where $\mathcal{N}(a, b)$ denotes a Gaussian distribution with mean $a$ and variance $b$, and $N(\cdot | a, b)$ will denote its density.

Bayesian (nonparametric) regression [30] consists of specifying a (nonparametric) prior distribution over the function $f$ and conditioning on the observations to obtain a posterior distribution on $f$. This prior is often chosen to be some (real-valued) stochastic process $g : \mathcal{X} \times \Omega \to \mathbb{R}$ where $(\Omega, \Sigma, \mathbb{P})$ is some probability space. These can be thought of as a random function since $\forall \omega \in \Omega$, $g(\cdot; \omega)$ will be a function, and $\forall x \in \mathcal{X}$, $g(x; \cdot)$ will be a random variable.

A common choice of prior is a GP [66]. A GP is fully determined by its mean function $\mu : \mathcal{X} \to \mathbb{R}$ and covariance function (or kernel) $k : \mathcal{X} \times \mathcal{X} \to \mathbb{R}$, and hence often denoted by $\mathcal{GP}(\mu, k)$. Due to conjugacy properties of the Gaussian distribution, the posterior $f$ after conditioning on $(X, y)$ is once again a GP with posterior mean $\tilde{\mu}(x) = \mu(x) + k_{x,X}(k_{X,X} + \sigma^2 I)^{-1}(y - \mu_X)$ and covariance $\tilde{k}(x, x') = k(x, x') - k_{x,X}(k_{X,X} + \sigma^2 I)^{-1}k_{X,x'}$, where $k_{X,X} := (k(x_i, x_j))_{i,j=1}^n$, $k_{x,X} := (k(x, x_1), \ldots, k(x, x_n))$, $k_{X,x} := k_{x,X}^\top$, $\mu_X := (\mu(x_1), \ldots, \mu(x_n))^\top$ and $I$ is the $n \times n$ identity matrix. The properties of GP posteriors are inherited from $\mu$ and $k$. Whenever these are continuous, the posterior mean will also be continuous. This is the case for the most common choices of kernels, such as the Gaussian or Matérn kernels with sufficient smoothness. In those cases, discontinuities would have to be inputted manually or inferred from data [13], making the modelling of functions with a large number of discontinuities a challenge for most GP models.

In this paper, we consider instead models based on tree structures. Popular examples include Bayesian CART [16], dynamic regression trees [78] and Mondrian forests [45]. We will focus on BART [10; 36] due to its strong empirical performance and theoretical results. BART is a model which consists of a combination of regression trees. A *regression tree* is any step function $g_{\mathcal{T},\beta} : \mathcal{X} \to \mathbb{R}$ with $K$ leaves/partitions: $g_{\mathcal{T},\beta}(x) := \sum_{k=1}^K \beta_k \mathbb{1}_{\chi_k}(x)$. Here, we denote by $\mathcal{T} := \{\chi_k\}_{k=1}^K$, where $\chi_k \subset \mathcal{X}$, the partition of the domain, and by $\beta := (\beta_1, \ldots, \beta_K)^\top \in \mathbb{R}^K$ the leaf values. The function $\mathbb{1}_{\chi_k} : \mathcal{X} \to \mathbb{R}$ is an indicator function taking value 1 whenever $x \in \chi_k$ and 0 otherwise.

Similarly to decision trees and random forest, a single tree comprising of a large number of leafs might overfit to the training data. As a result, it is common to use an ensemble of shallow trees. We will call $T$-*additive regression tree* any function which takes the form of a sum of regression trees: $g_{\mathcal{E},\mathcal{B}}(x) := \sum_{t=1}^T g_{\mathcal{T}_t, \beta_t}(x)$, where $\mathcal{E} := \{\mathcal{T}_t\}_{t=1}^T$ and $\mathcal{B} := \{\beta_t\}_{t=1}^T$. Finally, we call *Bayesian additive regression tree* (BART) any distribution on the family of $T$-additive regression trees. Such a distribution can be constructed by specifying a (prior) distribution on the partition $\mathcal{E}$ and leaf values $\mathcal{B}$. BART is hence a stochastic process whose sample space $\Omega$ consists of the product space of $K_t$-partitions of $\mathcal{X}$ and $\mathbb{R}^{K_t}$. For simplicity, BART is usually restricted to an approximation domain $\mathcal{X} = [0, 1]^d$, but this is not a requirement in full generality.

Denote by $p$ the density of this prior distribution. We will follow the majority of the BART literature [9; 10; 71] and use prior models which factorise in the following way: $p(\mathcal{E}, \mathcal{B}) := \prod_{t=1}^T p(\mathcal{T}_t) p(\beta_t | \mathcal{T}_t)$ and $p(\beta_t | \mathcal{T}_t) := \prod_{k=1}^{K_t} N(\beta_{t,k} | 0, 1/(16T))$, where $T$ is the number of trees and $K_t$ is the number of leaves in the $t^{\text{th}}$ tree. The construction of the distribution on partitions is usually itself done via a tree generating stochastic process; see further details in [10] (and a full description in Appendix A).

Given this prior $\mathbb{P}$ on $T$-additive trees, we can condition on $(X, y)$ to obtain a posterior $\mathbb{P}_n$ (with density $p_n$). We focus on regression with i.i.d. Gaussian noise, but several generalisations exist [64]. The corresponding posterior distribution on parameters will then imply a posterior distribution

on $T$-additive trees. BART is hence another Bayesian model which we can use to approximate the integrand, using for example the posterior mean: $g^n(x) = \mathbb{E}_{\mathbb{P}_n}[f(x)] = \mathbb{E}[f(x)|X,y] = \int_\Omega g_{\mathcal{E},\mathcal{B}}(x)p_n(\mathcal{E},\mathcal{B})d\mathcal{E}d\mathcal{B}$.

Unfortunately, this is not available in closed form and needs to be approximated. This is usually done using Markov chain Monte Carlo (MCMC) methods [10; 62] (although optimisation methods can also be used [34]). At each iteration $j$, the MCMC algorithm produces $T$ regression trees. More precisely, for the $t^{\text{th}}$ regression tree, the algorithm returns a set of leaf values $\{\beta_{t,k}^j\}_{k=1}^{K_{t,j}}$ and a partition of the domain $\{\chi_{t,k}^j\}_{k=1}^{K_{t,j}}$ where $K_{t,j}$ is the number of leaves for tree $t$ at iteration $j$. This gives a T-additive regression tree which we will denote $g_j^n$. After $m$ iterations, the MCMC mean is given by:

$$\hat{g}^n(x) = \tfrac{1}{m}\sum_{j=1}^m g_j^n(x) = \tfrac{1}{m}\sum_{j=1}^m \sum_{t=1}^T \sum_{k=1}^{K_{t,j}} \beta_{t,k}^j \mathbb{1}_{\chi_{t,k}^j}(x). \tag{2}$$

A similar expression can be obtained for the variance of our posterior on $f$ at $x \in \mathcal{X}$. However, when compared to GPs, the entire BART posterior can in fact provide us with a more complex, possibly multimodal, distribution on $f$.

# 3 Numerical Integration with Bayesian Additive Regression Trees

The Bayesian approach to numerical integration can be succinctly summarised in the following steps:

1. *Posit a Bayesian prior on the integrand $f$.* This consists of a stochastic process $g : \mathcal{X} \times \Omega \to \mathbb{R}$ together with a prior measure $\mathbb{P}$ on $\Omega$.
2. *Condition this prior on the data $(X, y)$ to get a posterior on $f$.* This is given in terms of the posterior measure $\mathbb{P}_n$, and the posterior mean is $g^n := \mathbb{E}_{\mathbb{P}_n}[g] = \mathbb{E}[g|X,y]$.
3. *Obtain the posterior distribution on $\Pi[f]$.* This is given by the marginal distribution obtained by integrating out $\Pi$, and a point estimate for the posterior mean $\Pi[f]$, given by $\hat{\Pi}_{\text{BPNI}}[f] = \Pi[g^n]$.

We remark that the posterior distribution on $\Pi[f]$, or even just the estimate $\hat{\Pi}_{\text{BPNI}}[f]$, may not be available in closed-form. In that case, MCMC estimates can be used (see the following section). In the case of GP-BQ with a prior $\mathcal{GP}(\mu, k)$, the posterior on $\Pi[f]$ is available in closed-form [4] and is given by a Gaussian with mean $\mathbb{E}[\Pi[f]|X,y] = \Pi[\mu] + \Pi[k_{\cdot,X}](k_{X,X} + \sigma^2 I)^{-1}(y - \mu_X)$ and variance $\mathbb{V}[\Pi[f]|X,y] = \Pi\bar{\Pi}[k] - \Pi[k_{\cdot,X}](k_{X,X} + \sigma^2 I)^{-1}\Pi[k_{X,\cdot}]$, with $\Pi[k_{\cdot,X}] = (\Pi[k(\cdot,x_1)], \ldots, \Pi[k(\cdot,x_n)])$, $\Pi[k_{X,\cdot}] = \Pi[k_{\cdot,X}]^\top$ and $\bar{\Pi}$ denotes the integration functional with respect to the second input. The posterior mean can be expressed as a quadrature rule whose weights depend on $X$, $k$ and $\sigma$. This expression will be available in closed form whenever $\Pi[k(\cdot,x)]$, known as the *kernel mean*, is available in closed form. This is potentially a restrictive condition, but these integrals can themselves be approximated using quadrature, or the kernels could be constructed with this condition in mind [53; 54].

A significant challenge for GP-BQ is the computational cost, which is $\mathcal{O}(n^3)$ due to the need to invert an $n \times n$ matrix. This can possibly be alleviated using fast GP algorithms, but often at the cost of introducing approximations to the posterior. The GP-BQ method is hence more expensive than most standard MC methods, but this should be balanced with significantly faster convergence rates; see [4; 41; 82]. Overall, this method should hence be preferred to MC methods whenever the number of quadrature points $n$ is small or moderate, or when the integrand is itself expensive.

We now derive a new BPNI method via BART. Suppose that $g$, $\mathbb{P}_0$ are a BART prior (as described in Sec. 2) and we have data $(X, y)$; the posterior on $f$ implies a posterior on $\Pi[f]$. An important distinction with GP-BQ is that the posterior on $\Pi[f]$ is not available in closed-form, and must be approximated. It is also not fully specified by the mean and variance, and may, in fact, be multi-modal. However, the mean of this posterior can be useful if a point estimate for $\Pi[f]$ is required, and the variance can be a useful summary of uncertainty. Expressions for both of these quantities are included in the proposition below for completeness.

**Proposition 1** (**MCMC for BART-Int**). *The MCMC approximation of $\Pi[f]$ consists of samples:*

$$s_j = \Pi\left[g_j^n\right] = \sum_{t=1}^T \sum_{k=1}^{K_{t,j}} \beta_{t,k}^j \Pi\left[\mathbb{1}_{\chi_{t,k}^j}\right], \qquad j \in \{1, \ldots, m\}.$$

*These lead to estimates of the mean* $\widehat{\mathbb{E}}_m[\Pi[f]|X,y] = \Pi[\hat{g}^n] = \frac{1}{m}\sum_{j=1}^{m} s_j$ *and variance* $\widehat{\mathbb{V}}_m[\Pi[f]|X,y] = \frac{1}{(m-1)}\sum_{j=1}^{m}(s_j - \Pi[\hat{g}^n])^2.$

The posterior samples $\{s_j\}_{j=1}^{m}$ can only be obtained in a closed form whenever probabilities of the form $\Pi(\chi)$ can be computed for any $\chi$ in the partition. This issue corresponds to the well-known issue of intractable kernel means for GP-BQ. The simplest case for which $\Pi(\chi)$ can be computed is when $\Pi$ is the uniform distribution. If $\pi$ is available in closed form, one option is to model $f\pi$ as the integrand, in which case the integral is once again against a uniform. However, this makes the specification of a BART prior challenging since very little will usually be known about $f\pi$, and as a result the uncertainty quantification may be unreliable.

Another solution, used in this paper and commonly used for intractable kernel means, is to approximate these probabilities using samples $\{\tilde{x}_i\}_{i=1}^{l}$ representative of $\Pi$. This leads to approximate MCMC samples and an estimate of $\Pi[f]$ of the form:

$$\hat{s}_j^l = \frac{1}{l}\sum_{i=1}^{l} g_j^n(\tilde{x}_i) = \frac{1}{l}\sum_{i=1}^{l}\sum_{t=1}^{T}\sum_{k=1}^{K_{t,j}}\beta_{t,k}^j \mathbb{1}_{\chi_{t,k}^j}(\tilde{x}_i),$$

$$\widehat{\mathbb{E}}_{\mathrm{m}}^l[\Pi[f]|X,y] = \frac{1}{m}\sum_{j=1}^{m}\hat{s}_j^l. \qquad \widehat{\mathbb{V}}_{\mathrm{m}}^l[\Pi[f]|X,y] = \frac{1}{(m-1)}\sum_{j=1}^{m}\left(\hat{s}_j^l - \left(\frac{1}{m}\sum_{k=1}^{m}\hat{s}_k^l\right)\right)^2.$$

Using this approach might be counter-intuitive since it means that some of the uncertainty is not quantified in a Bayesian manner. However, when the integrand $f$ is expensive but a very large number of data points are available (i.e. $n \ll l$), the additional error can be made to be negligible relative to the overall integration error. These settings are common in practice, as will be shown in Sec. 5.

A significant advantage of BART-Int over GP-BQ is the computational cost. For BART, the cost is $\mathcal{O}(Tmn)$ (or $\mathcal{O}(Tm(n+l))$), although the constant depends on the properties of the trees (see [63] for a detailed analysis). When $n$ is large, this can be much cheaper than the $\mathcal{O}(n^3)$ for GP-BQ. We note that $\{x_i\}_{i=1}^{n}$ can be selected through any quadrature rule. However, when $f$ is computationally expensive, it may be preferable to adaptively select $\{x_i\}_{i=1}^{n}$ using tools from experimental design and active learning [75]. Recent advances in this direction in numerical integration focus on minimising some distance between $\Pi$ and $\{x_i\}_{i=1}^{n}$ (see e.g. kernel herding [8], Stein Variational Gradient Descent [49] and Stein points [7]). All of these methods could be combined with our BART-Int estimators, but such an approach may be sub-optimal in the sense that our objective is not to approximate $\Pi$, but only $\Pi[f]$. We propose instead a method which focuses on improving the fit of the BART posterior mean. Our approach is hence closer to previous active learning strategies for GP-BQ, including the work of [3; 31]. It is summarised in Algorithm 1.

---

**Algorithm 1** Sequential Design for BART-Int

1: **Inputs:** Initial point(s) $X$ (set of size $n_{\mathrm{ini}}$), response(s) $y$, number of candidate design points $S$, total number of points $n$, acquisition criterion $J$ and number of MCMC samples $m$.
2: **for** $i \in \{n_{\mathrm{ini}}, \ldots, n-1\}$ iterations **do**
3:      Obtain $m$ posterior samples (given $X^i, y^i$) & pick a candidate set $\mathcal{C} = \{c_1, \ldots, c_S\} \subset \mathcal{X}$.
4:      Find $c^* = \mathrm{argmax}_{c \in \mathcal{C}} J(c)$ & set $X^{i+1} \leftarrow X^i \cup \{c^*\}$, $y^{i+1} \leftarrow y^i \cup \{y_c^*\}$, $y_c^* = f(c^*) + \epsilon^*$.
5: **end for**

---

The second step consists of comparing the suitability of each point in $\mathcal{C}$ according to some criterion $J : \mathcal{X} \to \mathbb{R}$, then adding the best point to our design set. At iteration $i$, we propose to use

$$J(c) = \widehat{\mathbb{V}}_m[f(c)\pi(c)|X^i, y^i], \tag{3}$$

which leads to selecting points where the uncertainty in $f$ is highest. Alternatively, one could choose $c$ which minimises the posterior variance: $J(c) = -\widehat{\mathbb{V}}_m[\Pi[f]|X^i \cup \{c\}, y^i \cup \{y_c\}]$. However, this would require training a different MCMC sampler for each point in $\mathcal{C}$, which would significantly increase the computational cost.

## 4 Theoretical Results

We now introduce novel concentration results. All proofs are given in Appendix C. These results hold for BART-Int, but are also of independent interest since they can lead to rates for other BPNI

methods. The results will hold for an integrand in a normed subspace $\mathcal{H}$ of $L^2(\Pi) := \{h : \mathcal{X} \to \mathbb{R} \text{ s.t. } \|h\|_{L^2(\mathcal{X})}^2 := \int_{\mathcal{X}} h^2(x)\pi(x)dx < \infty\}$. They will depend on the contraction rate of the posterior onto the integrand, as measured in some empirical norm: $\|f\|_n = (\frac{1}{n}\sum_{i=1}^n f(x_i)^2)^{1/2}$. Below, we will use $(X^n, y^n)$ to denote a dataset $(X, y)$ of size $n$.

**Theorem 1** (**Concentration Bound for BPNI**). *Suppose we have a normed space $\mathcal{H} \subseteq L^2(\Pi)$ such that the integrand $f \in \mathcal{H}$. Furthermore, suppose that our BPNI posterior is based on a stochastic process $g$ (jointly measurable over $\mathcal{X} \times \Omega$) and a sequence of data $\{(X^n, y^n)\}_{n \in \mathbb{N}}$ such that $\forall \omega \in \Omega, g(\cdot, \omega) \in \mathcal{H}$ and $\exists N \in \mathbb{N}_+$ such that:*

*A1. $\exists \{\varepsilon_n\}_{n \geq N}$ such that $\lim_{n \to \infty} \mathbb{P}_n[\|f - g\|_n > A_n \varepsilon_n] = 0$ for any $A_n \to \infty$ as $n \to \infty$.*

*A2. $\exists \{\gamma_n\}_{n \geq N}$ with $\gamma_n \to 0$ as $n \to \infty$ such that $\sup_{\|h\|_{\mathcal{H}} \leq 1} |\frac{1}{n}\sum_{i=1}^n h(x_i) - \Pi[h]| = \mathcal{O}(\gamma_n)$.*

*Then, we have $\lim_{n \to \infty} \mathbb{P}_n[|\Pi[f] - \Pi[g]| > C_n \max(\varepsilon_n, \gamma_n)] = 0$ for any $C_n \to \infty$ as $n \to \infty$.*

We now make a number of remarks on the assumptions and statement of the theorem:

1. To guarantee concentration of a BPNI posterior, we require two ingredients: (i) the Bayesian posterior should concentrate on $f$ as $n$ grows, and (ii) the sequence of point sets $X^n$ should give a sequence of quadrature rules which can integrate both $f$ and $g^n$ for any $n$ large enough. The rates at which (i) and (ii) occur then control the overall concentration rate of the BPNI posterior.

2. The assumptions in this theorem are stated in a way such that the order in $n$ at which concentration of the BPNI posterior occurs is explicit. In particular, A1 is a standard assumption in Bayesian nonparametric and guarantees concentration of the posterior on $f$ at rate controlled by $\mathcal{O}(\varepsilon_n)$. Similarly, A2 is a common assumption for quadrature rules and guarantees a quadrature rate of $\mathcal{O}(\gamma_n)$. The resulting concentration rate of the BPNI posterior is then controlled by $\mathcal{O}(\max(\varepsilon_n, \gamma_n))$, the slowest of these two rates.

3. A number of quadrature rules can integrate functions in $L^2(\Pi)$ at a rate $\mathcal{O}(n^{-1/2})$ (i.e. $\gamma_n = n^{-1/2}$), but have faster worst-case integration rates for subspaces of $L^2(\Pi)$ which are sufficiently regular. For this reason, we introduced the subspace $\mathcal{H}$ and the conditions that $f \in \mathcal{H}$ and $\forall \omega \in \Omega$, $g(\cdot, \omega) \in \mathcal{H}$, in order to allow for faster rates for $\gamma_n$.

4. Results of the form in A1 are available for a variety of Bayesian nonparametric models [30], including GPs [80] and Bayesian neural networks [61]. This is a rapidly evolving field and most results were only derived in recent years. The advantage of Thm. 1 is that any contraction result derived in the future can be plugged in to understand the corresponding BPNI method.

5. Assumption A2 will hold for most reasonable choices of design points. For example, it would hold for $V$-uniformly ergodic MCMC when $\mathcal{H} = \{h \in L^2(\Pi) : \sup_{x \in \mathcal{X}} |h(x)|/V(x) < \infty\}$ (under regularity conditions given in [69]). Note that Assumption A1 may also depend on the design points, and a discussion of various cases can be found in [39].

6. As for BART-Int, we might sometimes need to approximate the BPNI using samples $\{\tilde{x}_i\}_{i=1}^l$. In that case, it is sufficient for $\{\tilde{x}_i\}_{i=1}^l$ to satisfy an assumption of the form of A2 for Thm. 1 to hold. In general, it will be possible to take $l$ much larger than $n$, so we could potentially even allow for slower quadrature rates than $\mathcal{O}(\gamma_n)$.

Now that we have discussed our result on concentration of BPNI posteriors, we consider implications for BPNI with tree-based models. Most results are presented for the space of $\alpha$-Hölder continuous functions, denoted $\mathcal{H}^\alpha$, where $\|f\|_{\mathcal{H}^\alpha} := \sup_{x,y \in \mathcal{X}} |f(x) - f(y)|/\|x - y\|_2^\alpha < \infty$ (when $\alpha = 1$, we recover Lipschitz functions). Several results are of interest:

- [71; 72] showed that Bayesian CART and BART concentrate on targets which are sums of functions in $\mathcal{H}^\alpha$ ($0 < \alpha \leq 1$) that are constant in $d - d_0$ coordinates at a rate $\varepsilon_n = n^{-\frac{\alpha}{2\alpha + d_0}} \log^{\frac{1}{2}} n$. This does not require a-priori knowledge of the coordinates in which the function is constant.

- [48] showed that the soft-BART posterior raised to a fractional power concentrates at the rate $\varepsilon_n = n^{-\frac{\alpha}{2\alpha + d_0}} \log^\beta n + \sqrt{n^{-1} d_0 \log d}$ for any $\beta \geq \alpha(d_0 + 1)/(2\alpha + d_0)$, even when $\alpha > 1$.

These results are not directly comparable with the most of the literature for GP-BQ [4; 41; 42] since these consider the interpolation setting (where $y_i = f(x_i)$). The closest result is in [82], which

presents a result for regression with i.i.d. Gaussian noise and convergence in expectation. The rate in [82] is identical to those above except for for sparse high-dimensional functions where it is slower.

Recent Bernstein-von Mises results for BART presented in [70] (Thm. 5.2) also provide guarantees on the uncertainty quantification provided by the BART-Int posterior distribution on $\Pi[f]$, and show that in the asymptotic limit of $n \to \infty$, the Bayesian credible intervals will coincide with frequentist confidence intervals. Similar results were obtained for sparse deep networks in [81].

Before concluding, we note that any MCMC approximation required to approximate the posterior will also impact the convergence. This is made more precise through the following result.

**Proposition 2** (**MCMC Approximation for BPNI**). *Let $\hat{\mathbb{E}}_m[\Pi[g]]$ denote the MCMC approximation of $\mathbb{E}[\Pi[g]]$ where the expectation is with respect to some measure $\mathbb{P}$. Assume $\Pi[g] \in L^2(\mathbb{P})$, where the Markov chain targets $\mathbb{P}$ and is geometrically ergodic and reversible. Then, $\exists \sigma_{MCMC} > 0$ such that*

$$\sqrt{m}|\mathbb{E}[\Pi[g]] - \hat{\mathbb{E}}_m[\Pi[g]]| \to \mathcal{N}(0, \sigma_{MCMC}^2)$$

*in distribution as $m \to \infty$.*

Geometric ergodicity is a well-studied concept in MCMC theory; see Appendix C.3. Combining Thm. 1 and Prop. 2 gives us some intuition as to how to balance the computational cost of increasing $n$ and $m$ in order to obtain the smallest possible approximation error. Unfortunately, to the best of our knowledge, the ergodicity of MCMC samplers commonly used for BART has not yet been studied.

## 5 Numerical Experiments

We now illustrate BART-Int and GP-BQ on a range of synthetic problems and a Bayesian survey design problem. We emphasise that our goal is to compare methods which can provide a Bayesian quantification of uncertainty; as a result, it is very much possible that non-Bayesian methods could have better point estimate performance on some of these problems. For BART-Int, we used the default prior settings in dbarts [20], whereas for GP-BQ we used a Matérn kernel whose lengthscale was chosen through maximum likelihood. Further details can be found in Appendix D.

| Setup | Method | Cont | Copeak | Disc | Gaussian | Oscil | Prpeak | Step |
|---|---|---|---|---|---|---|---|---|
| $d = 1$ | BART-Int | 1.21 | **7.32e-01** | **6.06e-02** | **6.80e-01** | **228** | 2.58 | **7.85e-03** |
| $n_{\text{ini}} = 20$ | MI | **7.11e-01** | 8.99e-01 | 1.55e-01 | 8.41e-01 | 251 | 1.28 | 5.90e-02 |
| $n_{\text{seq}} = 20$ | GP-BQ | 9.38e-01 | 3.41 | 1.61e-01 | 7.42e-01 | 229 | **9.88e-01** | 2.88e-02 |
| $d = 10$ | BART-Int | **9.10e-04** | 43.8 | **1.31e-02** | 3.37e-03 | 1.19e-02 | 1.77e-02 | **4.33e-03** |
| $n_{\text{ini}} = 200$ | MI | 2.78e-03 | **2.12e-01** | 1.33e-01 | 1.18e-02 | 8.43e-02 | 3.70e-03 | 2.46e-02 |
| $n_{\text{seq}} = 200$ | GP-BQ | 1.52e-03 | 2.79e-01 | 1.84e-01 | **3.02e-03** | **2.39e-03** | **6.59e-04** | 8.36e-03 |

Table 1: Integration of the six Genz test functions and a step function. The values correspond to the mean absolute percentage error (MAPE) over 20 separate runs.

**Genz Functions**  We start with a standard benchmark set of multi-dimensional integrands proposed by Genz [28] (see Appendix D). This consists of six functions in $\mathcal{X} = [0, 1]^d$, where $d$ can be varied to adjust the level of difficulty of the problem. We also add a step function: $f_{\text{step}}(x) = \mathbb{1}_{\{x_1 \in (0.5, 1]\}}(x)$, where $x_1$ is the first component of $x$. We use $n_{\text{ini}} = 20d$ design points, with $n_{\text{seq}} = 20d$ additional points selected by the sequential scheme in Algorithm 1.

Table 1 shows the mean absolute percentage error (MAPE) of BART-Int ($m = 1500$, $T = 200$ $m = 1000$, $T = 50$, with a burn-in of 1000 and keeping every 5 samples afterwards), GP-BQ and Monte Carlo Integration (MI) for all six Genz functions in $d = 1$ and 10. The MAPE is given by given by $\frac{1}{r}\sum_{t=1}^{r} |\Pi[f] - \hat{\Pi}_t[f]|/|\Pi[f]|$, where $\hat{\Pi}_t[f]$ for $t = 1, \ldots, r$, are estimates of $\Pi[f]$ for $r$ different initial i.i.d. uniform point sets.

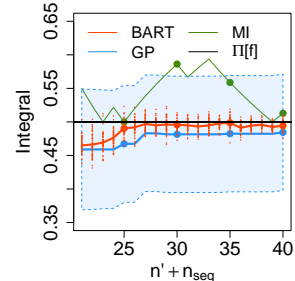

Figure 1: Integration of $f_{\text{step}}$ against a uniform, $d = 1$, $n_{\text{ini}} = 20$ and $n_{\text{seq}} = 20$.

BART-Int outperforms GP-BQ when $f$ is not continuous, e.g. the Step functions and the Discontinuous functions. In those cases, the posterior distribution of BART (red dots) is also more concentrated

around the truth than GP-BQ (whose 95% credible intervals are in shaded blue); see Fig 1. However, GP-BQ is strongest when estimating integrals of smooth continuous in $d = 1$ (see all other integrands). This is to be expected since these functions are smooth enough to be well modelled by the GP (see convergence rates in [82]), while the step-function nature of BART makes it more appropriate for non-smooth functions. Finally, in $d = 10$, BART-Int outperforms GP-BQ for all integrands as it is adaptive to the important features of the integrand, with the largest gains being once again for discontinuous functions.

**Computational Complexity**   The computational complexity of BART-Int is $\mathcal{O}(Tmn)$, which is much cheaper than the $\mathcal{O}(n^3)$ for GP-BQ with even moderately large $n$. This behaviour can be seen empirically in Figure 2, noting $T$ and $m$ have been fixed. The computational time of BART-Int is based on the assumption that the tree-based operations are constant, which is reasonable so long as the tree sizes are moderate [63]. Furthermore, it has been shown empirically that the run time of BART is almost independent of the dimension, $d$, of the data, see e.g. [63] and section 6 of [10]. However, with larger $d$, it would require a longer burn-in period for the parameter space to be fully explored, and thus the run-time for BART (thus also BART-Int) would increase for the purpose of approximation accuracy.

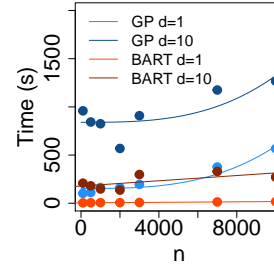

Figure 2: Run-times of GP-BQ & BART-Int for $\Pi[f_{\text{step}}]$ (w.o. seq. design).

**Rare Events Simulation**   Rare-event simulation is an important tool in risk management. It is used in a range of applications, such as for safety testing autonomous driving systems [59], queuing systems [73] and finance [6].    Traditionally, rare-event simulation is tackled using Monte Carlo methods [73], but we propose to use BART-Int instead.

Let $\ell : \mathbb{R}^d \to \mathbb{R}$ be a measurable function representing our loss. The central quantity of interest here is the probability of obtaining a loss larger than some constant $\gamma \in \mathbb{R}$, which can be written as $p_\gamma = \int_{\mathcal{X}} \mathbb{1}_{\{\ell(x) > \gamma\}}(x)\Pi(dx)$.

We consider a problem of high-dimensional portfolio management adapted from [6]. Suppose we have $d$ loans to obligors, each with value $c_i$ for $i = 1, \ldots, d$. Let $x_i$ denote the financial strain on loan $i$, and suppose $d_i$ is a thresholds after which default occurs. We assume that the distribution of financial strains is $\text{Exp}(1)$. We can define the portfolio loss as $\ell(x) = \sum_{i=1}^{d} c_i \mathbb{1}_{\{x_i > d_i\}}(x)$. Given a threshold $\gamma$, we can compute the probability of making a loss greater than $\gamma$ as $p_\gamma$.

|  | Method | MAPE | Std. Err. |
|---|---|---|---|
| | BART-Int | 1.71e-01 | 2.56e-02 |
| $d = 5$ | MI | 1.95e-01 | 2.29e-02 |
| $n = 2500$ | GP-BQ | **1.68e-01** | 2.09e-02 |
| | BART-Int | **1.56e-02** | 2.35e-03 |
| $d = 10$ | MI | 9.98e-01 | 4.47e-04 |
| $n = 5000$ | GP-BQ | 2.72e-02 | 5.20e-03 |
| | BART-Int | **8.40e-03** | 1.60e-03 |
| $d = 20$ | MI | 9.94e-01 | 6.34e-04 |
| $n = 10000$ | GP-BQ | 2.92e-02 | 4.90e-03 |

Table 2: Performance on the portfolio loss event over 20 runs.

We set $\gamma = 2$, $d_i = 0.5i$ and $c_i = 0.2i$. We use the same experimental settings as for the Genz functions, but do not use a sequential design and instead set $n = 500d$. The number of post-burn-in samples is chosen to be $10^4$. We can see that for $d = 5$ most of the algorithms perform similarly, but as $d$ increases BART-Int gradually performs better, especially when compared to GP-BQ.

**Bayesian Survey Design**   In surveys, response variables are collected from a subset of a population based on answers to a set of questions. Survey design concerns sampling strategies to obtain population-representative estimates from these samples. The standard approach is simple or stratified random (Monte Carlo) sampling. Bayesian hierarchical models are also often used in this setting to analyse survey data in order to stabilise estimates and to borrow strength when making sub-population estimates for underrepresented subgroups or locations [27].

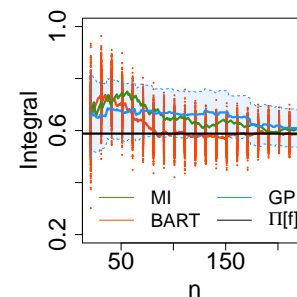

Figure 3: Proportion of individuals with log-salary >10.

We propose a new approach we term "Bayesian Survey Design", using BART-Int to adaptively choose the next individual to survey. Assume we have access to a small set of $n_{\text{ini}}$ survey responses with a mixture of continuous and categorical covariates (educational

attainment, age, etc). In addition, we assume there is a much larger set of $S$ individuals for whom demographic variables are known but the response variable is unknown but available for surveying. Such active surveying has previously also been explored in Bayesian active learning [26]. To demonstrate this approach, we use individual-level anonymised census data from the United States [79] and estimate the proportion of the population whose income is higher than around \$22,000, or log-income greater than 10. This is equivalent to computing the integral of the following indicator function: $f(x) = \mathbb{1}_{\{\text{log-income} > 10\}}(x)$ where the input consists of all other covariates in the survey and $\Pi$ is the distribution of these covariates in the population.

We consider a universe of $454,816$ possible respondents with 8 demographic covariates. After a one-hot encoding of categorical variables, the dimensionality is $d = 24$. We randomly select our initial set (of size $n_{\text{ini}} = 20$) and candidate set (of size $S = 10,000$). We compute ground truth using all $454,816$ observations and use Algorithm 1 to select $n_{\text{seq}} = 200$ new individuals to survey. As seen in Table 3 and Figure 3, BART-Int outperforms both MI

| Method | MAPE | Std. Err. |
|---------|----------|-----------|
| BART-Int | **4.66e-02** | 7.38e-03 |
| MI | 5.68e-02 | 7.01e-03 |
| GP-BQ | 9.73e-02 | 1.44e-02 |

Table 3: Performance on the Survey Design problem over 20 runs.

and GP-BQ. As expected, the posterior distributions contract to the population mean as $n$ increases. Furthermore, the BART-Int posterior (red dots) is centered around the truth, whereas the 95% credible intervals (blue shaded region) for GP-BQ seem overconfident at the wrong value around $n = 100$.

## 6 Related Work

Before concluding, we briefly comment on the connections between BART-Int and other tree-based algorithms for integration. Our work is most closely related to GP-BQ, but with tree-based priors. Although trees had not been considered in this literature, the nearest-neighbors method in [50] uses GP kernels which depends on indicator functions at Voronoi partitions, and yield approximations to $\Pi[f]$ which closely resembles that in our paper.

We note that tree-based methods have received significant interest in the literature in numerical integration, but previous work does not come with a Bayesian interpretation. For example, the VEGAS algorithm [47] from the physics literature can be thought of as a tree-based method to reduce the variance of MI. More recent work also includes MCMC and SMC samplers with proposals estimated using trees [22; 33]. Finally, [24] also uses regression trees as surrogates, but again in a non-Bayesian framework.

## 7 Discussion

We proposed a novel BPNI algorithm which uses BART instead of a GP to model the integrand. BART has several advantages in settings where GPs do not perform well. It is easy to tune, robust to overfitting and enjoys a low computational complexity. It is also a natural model for discontinuous functions and can automatically adapt to sparse high-dimensional functions. We demonstrated these advantages through theoretical results, including contraction of the posterior distribution onto the value of the integral, and through a set of benchmark tests and a survey design experiment. However, it is also important to note that BART-Int did not perform as well as GP-BQ in certain low dimensional and smooth settings. We therefore see BART-Int as a useful addition to the toolset of BPNI which complements, rather than replaces, GP-BQ.

There are a number of potential directions for future research. From an applications-viewpoint, there are interesting parallels to be made with algorithms used to estimate treatment effects in causal inference, and we foresee some possible use for BART-Int in this field. This was discussed in more details in an opinion piece published in [32]. In terms of theory, little is known about the ergodicity of the MCMC sampler used to sample posterior trees. Further work in this area could help practitioners understand the impact of the MCMC approximation on the estimate of the integral for BART-Int. In terms of methodology, while the sequential design approach proposed in this paper uses the (estimated) posterior variance to select new query points, it would be interesting to explore how the use of other acquisition functions may improve the estimates. Similarly, extensions of BART, such as soft-BART [48] and heteroscedastic-BART [64], should also be of practical interest, as they may provide additional advantages over GP-based methods. Finally, some tree-based models can naturally handle categorical variables without the need for one-hot encoding; this could significantly improve the performance for problems such as the Bayesian survey design experiment in this paper.

## Broader Impact

Our paper is about numerical integration, a common computational problem in machine learning. It provides an approach to improve the accuracy of such approximations, as well as obtaining credible intervals representing our uncertainty about the value of the integral. The increased accuracy of the method may allow the users to reduce their computational requirements, which may, further down the line, have some impact on mitigating the impact of large computer clusters on climate change.

However, the broader impact of the method will mostly depend on the applications of the algorithms that it will be used to enhance: applications to ethical algorithms will have a positive impact, but application to unethical algorithms will also have a negative impact. For example, in the Bayesian survey design problem studied in the numerical experiments section, we have shown that there is potential for application in predicting population proportions resulting from binary outcome variables. The categorical variables for instance, however, may be unethically used as explanatory variables for certain problems, and this would be an issue depending on how the conclusions are drawn.

## Acknowledgments and Disclosure of Funding

The authors would like to thank Matthew Fisher, Takuo Matsubara and Chris Oates for useful feedback on a previous draft of the paper. HZ, XL were supported by the EPSRC Centre for Doctoral Training in Modern Statistics and Statistical Machine Learning (EP/S023151/1) and the Department of Mathematics of Imperial College London. HZ was supported by Cervest Limited. XL was supported by the President's PhD Scholarships of Imperial College London. RK and ZS were supported by the Undergraduate Research Opportunities Programme of Imperial College London. FXB was supported by the Lloyd's Register Foundation Programme on Data-Centric Engineering and the Alan Turing Institute under the EPSRC grant [EP/N510129/1], and through an Amazon Research Award on "Transfer Learning for Numerical Integration in Expensive Machine Learning Systems".

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
