[Supplementary Material]

# Appendices

This supplementary material is separated into four sections. First, in Appendix A, we provided a detailed description of the BART prior considered in the paper. Secondly, in Appendix B we give a concise description of the posterior sampling procedure. Then, in Appendix C, we provide proofs for the theoretical result in the main text. Finally, in Appendix D, we provide additional details on the experimental setting as well as additional numerical results.

## A  The Bayesian Additive Regression Trees Prior

Following our definition in Section 3, for a fixed number of regression trees $T$, a *T-additive regression tree* $g_{\mathcal{E},\mathcal{B}}$ is essentially determined by $(\mathcal{T}_1, \beta_1), \ldots, (\mathcal{T}_T, \beta_T)$, where for each tree $t$, $\mathcal{T}_t$ is a partition of $\mathcal{X}$ into $K_t$ subsets and $\beta_t$ is a vector of leaf values. Additionally, $\sigma$ is the standard deviance of the Gaussian observational noise. This section summarizes the discussion in [10] on how the prior $\mathbb{P}((\mathcal{T}_1, \beta_1), \ldots, (\mathcal{T}_T, \beta_T), \sigma)$ is chosen. An appropriate prior can effectively prevent the individual trees from being overly influential, thus regularizing the fit as an ensemble model. We refer the reader to [10] for more details. Details about our specific choice of hyper-parameters for the priors can be found in Section D.1.

We will assume independence amongst tree components $(\mathcal{T}_t, \beta_t)$ and $\sigma$, and amongst each tree's terminal node parameters $\beta_t$. The distribution of the sum-of-trees model can hence be simplified as:

$$p((\mathcal{T}_1, \beta_1), \ldots, (\mathcal{T}_T, \beta_T), \sigma) = p(\sigma) \prod_t p(\beta_t | \mathcal{T}_t) p(\mathcal{T}_t)$$
$$p(\beta_t | \mathcal{T}_t) = \prod_k p(\beta_{t,k} | \mathcal{T}_t),$$

where $\beta_{t,k}$ is the $k$-th terminal node of the $t$-th tree. We further assume an identical form for each of the component $p(\mathcal{T}_t)$ and for $p(\beta_t | \mathcal{T}_t)$. The prior distributions to be specified are therefore $p(\mathcal{T}_t)$, $p(\beta_t | \mathcal{T}_t)$ and $p(\sigma)$.

**The $\mathcal{T}_t$ Prior**   The trees in BART are $k$-d trees and have axis-aligned splits. For the prior on $\mathcal{T}_t$, we will use what is commonly known as the "uniform" prior in the literature. This is usually specified in the form of a generative model which consists of three components. Firstly, we specify the probability that a given node of depth $l \in \mathbb{N} \cup \{0\}$ is terminal. This takes the form $p_{\text{split}} = \alpha(1 + l)^{-\beta}$, where $\alpha \in (0, 1)$ and $\beta \in [0, \infty)$. Secondly, a uniform distribution on $\{1, \ldots, d\}$ is used to decide which of the available variables $x_1, \ldots, x_d$ to split on at each interior node. Lastly, a uniform distribution on the set of possible values of that variable is used for the splitting rule assignment. See [9; 20] for more details. Other priors are also possible, such as the Galton-Watson prior [71].

**The $\beta_t | \mathcal{T}_t$ Prior**   For the distribution of leaf values given a tree, we use a Gaussian distribution:

$$p(\beta_t | \mathcal{T}_t) = \prod_{k=1}^{K_t} N\left(\beta_{t,k}; 0, \sigma_\beta^2\right).$$

Let $y_{\min} = \min_i y_i^n$ and $y_{\max} = \max_i y_i^n$. In [10], the authors suggested to rescale $y$ to have zero mean and to ensure that $y_{\min} = -0.5$ and $y_{\max} = 0.5$, then taking $\sigma_\beta = 0.25/\sqrt{T}$. This ensures that the prior on $g(x)$ is within the interval $(y_{\min}, y_{\max})$ with high probability.

**The $\sigma$ Prior**   For the prior on $\sigma$, we use the inverse chi-square distribution $\sigma^2 \sim \nu\lambda/\chi_\nu^2$. This is also a conjugate prior which, again, reduces the required computational effort in the MCMC procedure, as elaborated in [10]. To find the appropriate hyper-parameters, we introduce $q \in (0, 1)$, take $\hat{\sigma}$ of $\sigma$ as the sample standard deviation of $y^n$ and calibrate for $\nu$ and $\lambda$ such that $\mathbb{P}(\sigma < \hat{\sigma}) = q$. The authors of [10] recommended $(\nu, q) = (3, 0.90)$ as it tends to avoid extremes and we hence follow these recommendations.

**Number of Trees $T$**   Finally, the number of regression trees $T$ can either be chosen to be the default value $T = 200$, or through cross-validation. [10] further pointed out that, in general, the model performs reasonably well on prediction tasks so long as the value of $T$ is not too small.

# B  Bayesian Additive Regression Trees Posterior Sampling Procedure

In this appendix, we now describe the Bayesian backfitting MCMC algorithm first introduced in [10], and which is used throughout our experiments. We will make use of the notation: $\mathcal{T}_{-j} = \mathcal{E} \setminus \{\mathcal{T}_j\}$ and similarly $\beta_{-j} = \mathcal{B} \setminus \{\beta_j\}$ for $j = 1, \ldots, T$. Our target posterior is the distribution $\mathcal{E}, \mathcal{B}, \sigma | y^n$. To sample from this posterior, we will make use of a Gibbs sampler, which, at each iteration, draws $(\mathcal{T}_j, \beta_j) | (\mathcal{T}_{-j}, \beta_{-j}, \sigma, y^n)$ sequentially for $t = 1, \ldots, T$, then draws $\sigma$ from $\sigma | (\mathcal{E}, \mathcal{B}, y^n)$.

**Sampling individual trees**   We note that drawing from $(\mathcal{T}_j, \beta_j) | (\mathcal{T}_{-j}, \beta_{-j}, \sigma, y^n)$ is equivalent to drawing from

$$(\mathcal{T}_j, \beta_j) | R_j, \sigma, \qquad \text{where} \qquad (R_j)_k = y_k - \sum_{t \neq j} g_{\mathcal{T}_t, \beta_t}(x_k),$$

for $k = 1, \ldots, n$, which is the partial residual obtained without the $j$-th tree. Drawing from this distribution is the same as drawing from the posterior of the residuals regression model with the $j$-th tree $R_j = g_{\mathcal{T}_j, \beta_j}(x) + \epsilon$, where $\epsilon \sim \mathcal{N}(0, \sigma^2)$. Since the prior distribution $p(\beta_j)$ and the likelihood $p(R_j | \beta_j, \mathcal{T}_j, \sigma)$ are Gaussian, the posterior distribution attains a closed form up to a normalising constant

$$p(\mathcal{T}_j | R_j, \sigma) \propto p(\mathcal{T}_j) \int_{\mathbb{R}^{K_j}} p(R_j | \beta_j, \mathcal{T}_j, \sigma) p(\beta_j | \mathcal{T}_j, \sigma) d\beta_j,$$

where $d\beta_j$ is the Lebesgue measure on $\mathbb{R}^{K_j}$ and $K_j$ is the number of leaf nodes for tree $j$. Therefore we can now sample $\mathcal{T}_j$ and $\beta_j$ via the following two-step procedure: first sample from $\mathcal{T}_j | R_j, \sigma$, then sample from $\beta_j | \mathcal{T}_j, R_j, \sigma$.

The draw of $\mathcal{T}_j | R_j, \sigma$ is done via a Metropolis-Hastings algorithm with the following proposal. Given the current tree, grow a terminal node with probability $0.25$, prune a pair of terminal nodes with probability $0.25$, change a non-terminal node's split rule with probability $0.40$, and finally swap a split rule between parent and child with probability $0.10$.

**Sampling the standard deviation of the observational noise**   To sample from $\sigma | (\mathcal{E}, \mathcal{B}, y^n)$, we simply draw from the inverse chi-squared distribution defined in Appendix A.

# C  Proofs of Theoretical Results

In this appendix, we present the proofs of our theoretical results. Section C.1 provides a proof of Proposition 1, Section C.2 provides a proof of Theorem 1, and Section C.3 provides a proof of Proposition 2.

## C.1  Proof of Proposition 1

*Proof.* We begin by deriving the integral of some arbitrary $T$-additive tree $g_{\mathcal{E}, \mathcal{B}} : \mathcal{X} \times \Omega \to \mathbb{R}$ against the probability measure $\Pi$ on $\mathcal{X}$. Fix $\omega \in \Omega$. Then, using linearity of integration, we get:

$$\Pi[g_{\mathcal{E}, \mathcal{B}}(\cdot, \omega)] = \int_{\mathcal{X}} g_{\mathcal{E}, \mathcal{B}}(x, \omega) d\Pi(x) = \int_{\mathcal{X}} \sum_{t=1}^{T} \sum_{k=1}^{K} \beta_{t,k} \mathbb{1}_{\chi_{t,k}}(x) d\Pi(x)$$
$$= \sum_{t=1}^{T} \sum_{k=1}^{K} \beta_{t,k} \int_{\mathcal{X}} \mathbb{1}_{\chi_{t,k}}(x) d\Pi(x) = \sum_{t=1}^{T} \sum_{k=1}^{K} \beta_{t,k} \Pi(\chi_{t,k}).$$

We can use this expression to derive the posterior mean and variance for $\Pi[g]$ given the data $X, y$:

$$\mathbb{E}[\Pi[g_{\mathcal{E}, \mathcal{B}}] | X, y] = \mathbb{E}\left[ \sum_{t=1}^{T} \sum_{k=1}^{K} \beta_{t,k} \Pi(\chi_{t,k}) \Big| X, y \right],$$
$$\mathbb{V}[\Pi[g_{\mathcal{E}, \mathcal{B}}] | X, y] = \mathbb{E}\left[ \left( \Pi[g_{\mathcal{E}, \mathcal{B}}] - \mathbb{E}\left[ \Pi[g_{\mathcal{E}, \mathcal{B}}] | X, y \right] \right)^2 \Big| X, y \right].$$

Using a U-statistic estimate of these quantities based on posterior samples for the parameters leads to our desired result. $\quad\square$

## C.2 Proof of Theorem 1

*Proof.* Fixing $\omega \in \Omega$ (i.e. fixing a realisation from the stochastic process) and starting with the triangle inequality, we can decouple the integration error into several terms depending on $g(\cdot, \omega)$:

$$|\Pi[f] - \Pi[g(\cdot, \omega)]|$$
$$= \left|\Pi[f] - \Pi[g(\cdot, \omega)] + \tfrac{1}{n}\sum_{i=1}^{n} g(x_i, \omega) - \tfrac{1}{n}\sum_{i=1}^{n} g(x_i, \omega) + \tfrac{1}{n}\sum_{i=1}^{n} f(x_i) - \tfrac{1}{n}\sum_{i=1}^{n} f(x_i)\right|$$
$$\leq \left|\Pi[f] - \tfrac{1}{n}\sum_{i=1}^{n} f(x_i)\right| + \left|\Pi[g(\cdot, \omega)] - \tfrac{1}{n}\sum_{i=1}^{n} g(x_i, \omega)\right|$$
$$+ \left|\tfrac{1}{n}\sum_{i=1}^{n} g(x_i, \omega) - \tfrac{1}{n}\sum_{i=1}^{n} f(x_i)\right|. \tag{4}$$

Note that by definition of the worst-case integration error in $\mathcal{H}$, we have that for any $h \in \mathcal{H}$:

$$\left|\Pi[h] - \tfrac{1}{n}\sum_{i=1}^{n} h(x_i)\right| \leq \|h\|_{\mathcal{H}} \times \sup_{\|h\|_{\mathcal{H}} \leq 1} \left|\Pi[h] - \tfrac{1}{n}\sum_{i=1}^{n} h(x_i)\right| \tag{5}$$

First, using Equation 5 in Equation 4, then using assumption A2 gives us that whenever $n \geq N$, $\exists B > 0$ such that:

$$|\Pi[f] - \Pi[g(\cdot, \omega)]| \leq B\|f\|_{\mathcal{H}}\gamma_n + B\|g(\cdot, \omega)\|_{\mathcal{H}}\gamma_n + \left|\tfrac{1}{n}\sum_{i=1}^{n} g(x_i, \omega) - \tfrac{1}{n}\sum_{i=1}^{n} f(x_i)\right|$$
$$= B(\|f\|_{\mathcal{H}} + \|g(\cdot, \omega)\|_{\mathcal{H}})\gamma_n + \left|\tfrac{1}{n}\sum_{i=1}^{n} g(x_i, \omega) - \tfrac{1}{n}\sum_{i=1}^{n} f(x_i)\right|. \tag{6}$$

To tackle the third term, we can use the Cauchy-Schwartz inequality, which states that $\forall u, v \in \mathbb{R}^n$, we have $(\sum_{i=1}^{n} u_i v_i)^2 \leq (\sum_{i=1}^{n} u_i^2)(\sum_{i=1}^{n} v_i^2)$. Taking $u_i = g(x_i, \omega) - f(x_i)$ and $v_i = 1$, we get:

$$\left(\sum_{i=1}^{n} g(x_i, \omega) - \sum_{i=1}^{n} f(x_i)\right)^2 \leq n \sum_{i=1}^{n} (g(x_i, \omega) - f(x_i))^2.$$

Multiplying both sides by $n^{-2}$ and taking square roots, we end up with:

$$\left|\tfrac{1}{n}\sum_{i=1}^{n} g(x_i, \omega) - \tfrac{1}{n}\sum_{i=1}^{n} f(x_i)\right| \leq \left(\tfrac{1}{n}\sum_{i=1}^{n}(g(x_i, \omega) - f(x_i))^2\right)^{\frac{1}{2}} = \|g(\cdot, \omega) - f\|_n. \tag{7}$$

Plugging in Equation 7 into Equation 6, we get:

$$|\Pi[f] - \Pi[g(\cdot, \omega)]| \leq B(\|f\|_{\mathcal{H}} + \|g(\cdot, \omega)\|_{\mathcal{H}})\gamma_n + \|g(\cdot, \omega) - f\|_n$$
$$\leq B(\|f\|_{\mathcal{H}} + \|g(\cdot, \omega)\|_{\mathcal{H}})\max(\varepsilon_n, \gamma_n) + \|g(\cdot, \omega) - f\|_n,$$

Using this inequality, we have that:

$$\mathbb{P}_n\left(|\Pi[f] - \Pi[g(\cdot, \omega)]| > C_n \max(\varepsilon_n, \gamma_n)\right)$$
$$\leq \mathbb{P}_n\left(B(\|f\|_{\mathcal{H}} + \|g(\cdot, \omega)\|_{\mathcal{H}})\max(\varepsilon_n, \gamma_n) + \|g(\cdot, \omega) - f\|_n \geq C_n \max(\varepsilon_n, \gamma_n)\right)$$
$$\leq \mathbb{P}_n\left(\|g - f\|_n \geq (C_n - B(\|f\|_{\mathcal{H}} + \|g(\cdot, \omega)\|_{\mathcal{H}}))\max(\varepsilon_n, \gamma_n)\right)$$
$$= \mathbb{P}_n\left(\|g - f\|_n \geq A_n \max(\varepsilon_n, \gamma_n)\right)$$
$$\leq \mathbb{P}_n\left(\|g - f\|_n \geq A_n \varepsilon_n\right), \tag{8}$$

where we have taken $A_n = C_n - B(\|f\|_{\mathcal{H}} + \|g(\cdot, \omega)\|_{\mathcal{H}})$. Clearly, since $C_n \to \infty$ as $n \to \infty$ and $B, \|f\|_{\mathcal{H}}, \|g(\cdot, \omega)\|_{\mathcal{H}} < \infty$, we must have $A_n \to \infty$ as $n \to \infty$. Hence, combining Assumption A1 and the upper bound of Equation 8, we have:

$$\lim_{n\to\infty} \mathbb{P}_n[|\Pi[f] - \Pi[g]| > C_n \max(\varepsilon_n, \gamma_n)] = 0$$

which concludes our proof.

$\square$

## C.3 Proof of Proposition 2 and Discussion

*Proof.* Theorem 25 of [69] states that if a Markov chain with stationary distribution $\mathbb{P}$ is reversible and geometrically ergodic, then a central limit theorem holds for any $h \in L^2(\mathbb{P})$. Taking $h = \Pi[g]$, which is in $L^2(\mathbb{P})$ by assumption, concludes the proof. $\square$

Geometric ergodicity is a well-studied concept in MCMC theory which ensures that the chain mixes at a fast rate; see [69], Section 3.4., for a discussion of sufficient conditions, and Section 5.2. for alternative sufficient conditions to obtain a CLT. Stronger results such as convergence almost surely or in probability could also be obtained using stronger conditions on the Markov chain; see for example Theorem 4 of [69] or Theorem 17.0.1 of [51]. Finally, all of the results aforementioned hold in the asymptotic setting where the number of MCMC samples $m \to \infty$. However, finite $m$ results could be obtained using concentration inequalities; see for example [60].

# D    Additional Numerical Experiments

In this appendix, we provide additional details on the numerical experiments in the paper including the Genz functions in Section D.1, the step function in Section D.2 and the Bayesian survey design experiment in Section D.3.

Figure 4 provides a summary of the algorithm. We first observe some data pairs $\{(x_i, y_i)\}_{i=1}^n$ and then obtain the posterior on , which can be approximated by $m$ MCMC samples. We then integrate each of the samples and then take the mean to obtain an estimate of the integral $\hat{\Pi}_{\text{BPNI}}[f]$.

Figure 4: Bayesian Probabilistic Numerical Integration using Bayesian Additive Regression Trees (BART-Int). Here, $\Pi$ could either be known or replaced with an estimate $\hat{\Pi}$.

Our code relies on `gpytorch` [25] for kernel hyper-parameter (lengthscale) tuning for the GP, and `dbarts` [20] as backend for implementing BART-Int. We use the Imperial College London High Performance Computing and the Department of Mathematics NextGen High Performance Computing servers to conduct our experiments. Our code is available on GitHub `https://github.com/ImperialCollegeLondon/BART-Int` and is subject to spontaneous maintenance.

## D.1    Genz Integrand Families

The Genz functions [28] were taken from `http://www.sfu.ca/~ssurjano`, and are presented in Table 4. They have two sets of parameters — $d$ "ineffective" parameters $u$ and $d$ "effective" parameters $a$ which vary the level of difficulty. We use the default setting of $u = (0.5, \ldots, 0.5)^\top$ and scale $a$ suitably as the dimension increases to ensure numerical stability. Specifically, this is done by bounding the $L_1$-norm of $a$ so that numerical stability is obtained (see [76] for details). As ground truth, we analytically compute the integrals for these Genz test functions, which are again given in Table 4. We compare the performance of BART-Int for each function and make comparisons with two baselines: Monte Carlo integration (MI) and GP-BQ. Following the literature, we choose $\Pi$ to be the uniform distribution on $[0, 1]^d$.

For BART-Int, we used an MCMC sampler described in Appendix B with a burn-in of 1000 samples and took 5000 samples afterwards. These were then thinned by keeping every 5 samples. This led to $m = 1000$. For the BART model, we used $T = 50$ trees, and the pair $(\alpha, \beta) = (0.95, 2)$ for the terminating probability (see Appendix A for further details). We set $\sigma = 0.1$ to calibrate its inverse-chi-squared prior [20] due to our knowledge that there is no observation noise, but keep it to be non-zero to preserve the statistical properties of BART. For the rest of the hyper-parameters, we used the default setting from `dbarts`. Note that we have applied very little tuning to the fitting of BART.

For GP-BQ, we used a prior mean $\mu(x) = 0$ and the Matérn kernel with smoothness $3/2$:

$$k(x, y) = \left(1 + \frac{\sqrt{3}\|x-y\|_2}{\rho}\right) \exp\left(-\frac{\sqrt{3}\|x-y\|_2}{\rho}\right),$$

where $\|\cdot\|_2$ is the Euclidean norm. The parameter $\rho$ is called the lengthscale, which was selected by maximising the marginal likelihood. To compute the kernel means, we used a MI estimate with $l = 10^6$ randomly sampled points from $\Pi$.

All of our results are presented in the main text. To complement these, we show the empirical distribution of the number of leaf nodes of the BART-Int method for each function in Figure 5.

Recall that we chose the hyper-parameter values of $(\alpha, \beta) = (0.95, 2)$ for the prior on trees, which guarantees that trees with $1, 2, 3, 4$ and $\geq 5$ terminal nodes receive prior probability of $0.05, 0.55, 0.28, 0.09$ and $0.03$ respectively. As we can see, the posterior distribution of number of leaves per tree varies across target functions, demonstrating that BART is able to adapt to the target function.

For fixed targets, we see very little difference between the distribution for $d = 1$ and $d = 10$. This is sub-optimal since, as mentioned in [72], the optimal number of leaves is $\mathcal{O}\left(n^{\frac{d}{2\alpha+d}}\right)$ where $\alpha$ is the Hölder smoothness of the target function. For fixed $\alpha$, this suggests we should take a larger number of leaves in larger dimensions. This suggests further improvements in performance could be obtained by adapting the prior distribution as a function of $d$. On the other hand, the small number of leaves may also be seen as an advantage from a computational viewpoint.

Figure 5: Histogram distribution of the number of leaf nodes, $K$ over $T = 50$ trees for the Continuous, Corner Peak, Discontinuous, Gaussian, Oscillatory and Product Peak Genz function in dimensions $d = 1, 10$.

## D.2 Step Function

For the step function, we conducted experiments when integrating the function against either a uniform or a truncated Gaussian distribution. It is clear that with $\Pi$ being the uniform measure, $\Pi[f] = 0.5$ for all dimensions. When $\Pi$ is a multivariate Gaussian distribution with mean $x = (0.5, \ldots, 0.5)^\top$ and identity variance matrix truncated to $[0, 1]^d$, the integral is $\Pi[f] = 0.5$ (by symmetry).

We first provide additional results when $\Pi$ is uniform. The performance of BART-Int in this case is presented in Section 5. Figure 6 illustrates the posterior estimates for both the step function and its integral with BART-Int and GP-BQ. We can see that the posterior distribution of the integral for BART is more concentrated around the value of the true integral than the GP. We can also see that the GP has trouble estimating the discontinuity at $x = 0.5$. Finally a disadvantage of BART-Int is that we see that uncertainty for both algorithms enlarge at areas where data is not observed, but for the end regions near 0 and 1 BART exhibits lower uncertainty due to its stepwise property. It is true that tree-based algorithms do not perform well for extrapolation tasks and this is quite evidently shown with the uncertainty intervals at the end points.

Furthermore, Figure 7 illustrates the sequentially selected design points for each method. As we can see, both the BART and GP methods adaptively select points in areas not covered by the initial design points, and where the uncertainty about $f$ is hence greatest.

As another toy example, we ran BART-Int, GP-BQ and MI on the step function integrated against a truncated Gaussian measure. We started with $n_{\text{ini}} = 20$ design points and selected sequentially $n_{\text{seq}} = 20$ points according to the scheme introduced in Algorithm 1. The experimental set-ups for BART-Int and GP-BQ were the same as in the previous experiment. The experiment was repeated

| Family | Integrand | Parameter | Integral |
|---|---|---|---|
| Continuous (cont) | $\exp\left(-\sum_{i=1}^{d} a_i\lvert x_i - u_i\rvert\right)$ | $a_i = \frac{150}{d^3}$ | $\Pi_{i=1}^{d}\frac{1}{a_i}\left(2 - \exp(-a_i u_i) - \exp(a_i(u_i - 1))\right)$ |
| Corner Peak (copeak) | $\left(1 + \sum_{i=1}^{d} a_i x_i\right)^{-(d+1)}$ | $a_i = \frac{600}{d^3}$ | $\left(\frac{1}{d!\Pi_{i=1}^{d} a_i}\right)\sum_{k=0}^{d}\sum_{\substack{I\subset\{1,\dots,d\}\\\lvert I\rvert=k}}(-1)^{k+d}\left(1 + \sum_{i=1}^{d} a_i - \sum_{j\in I} a_j\right)^{-1}$ |
| Discontinuous (disc) | $\begin{cases}0, & \text{if } x_i > u_i \text{ for } i = 1,\dots,\min(2,d)\\ \exp\left(\sum_{i=1}^{d} a_i x_i\right), & \text{otherwise}\end{cases}$ | $a_i = \frac{10}{d^3}$ | $\begin{cases}\Pi_{i=1}^{d}\frac{1}{a_i}\left(\exp(a_i\min(1, u_i)) - 1\right) & \text{if } d > 1\\ \frac{1}{a_1}\left(\exp(a_1 u_1) - 1\right) & \text{if } d = 1\end{cases}$ |
| Gaussian Peak (gaussian) | $\exp\left(-\sum_{i=1}^{d} a_i^2(x_i - u_i)^2\right)$ | $a_i = \frac{100}{d^2}$ | $\pi^{\frac{d}{2}}\Pi_{i=1}^{d}\frac{1}{a_i}\left(\Phi(\sqrt{2}a_i(1 - u_i)) - \Phi(-\sqrt{2}a_i u_i)\right)$ |
| Oscillatory (oscil) | $\cos\left(2\pi u_1 + \sum_{i=1}^{d} a_i x_i\right)$ | $a_i = \frac{110}{d^{5/2}}$ | $\frac{1}{\Pi_{i=1}^{d} a_i}\sum_{k=0}^{d}\sum_{\substack{I\subset\{1,\dots,d\}\\\lvert I\rvert=k}}(-1)^k h_d\left(2\pi u_1 + \sum_{i=1}^{d} a_i - \sum_{j\in I} a_j\right)$ where $h_d(\cdot) = \begin{cases}\sin(\cdot), & d \equiv 1 \mod 4\\ -\cos(\cdot), & d \equiv 2 \mod 4\\ -\sin(\cdot), & d \equiv 3 \mod 4\\ \cos(\cdot), & d \equiv 0 \mod 4\end{cases}$ |
| Product Peak (prpeak) | $\Pi_{i=1}^{d}(a_i^{-2} + (x_i - u_i)^2)^{-1}$ | $a_i = \frac{600}{d^3}$ | $\Pi_{i=1}^{d} a_i\left(\arctan(a_i(1 - u_i)) - \arctan(-a_i u_i)\right)$ |

Table 4: The Genz functions and their true integrals, defined with respect to a uniform measure on $[0,1]^d$. The default $u$-parameter is $u = (0.5,\dots,0.5)$ and the $a$-parameter is chosen such that the order of the integral values is comparable across dimension. Here, $\Phi(x) = \int_{-\infty}^{x}(2\pi)^{-1/2}\exp(-x^2/2)dx$ refers to the cumulative distribution function of a standard Gaussian random variable.

Figure 6: Integration of the step function against a uniform distribution over $[0, 1]$ with BART-Int and GP-BQ with $n = 20$ points. *Left:* The posterior distribution on $\Pi[f]$. *Middle:* The BART posterior distribution on $f$. Posterior samples for BART are plotted as red points. *Right:* The GP posterior distribution on $f$. The lines represent the posterior mean, the shaded areas give 95% credible regions for the GP.

Figure 7: Illustration of adaptive selection of design points through Algorithm 1 on the step function integrated against a uniform measure over $[0, 1]$ with $n_{\mathrm{ini}} = 20$ and $n_{\mathrm{seq}} = 20$.

Figure 8: Histogram distribution of the number of leaf nodes, $K$ over $T = 200$ trees for the step function in dimensions $d = 1$ and $d = 10$.

with 20 sets of initial points that were sampled randomly and independently from $\Pi$. Over these runs, BART-Int achieved the smallest MAPE of 1.81e-02, whereas MI gave 1.15e-01 and GP-BQ yielded 2.83e-02.

### D.3 Bayesian Survey Design

To process the dataset in our experiments, we first randomly selected $n_{\text{ini}} = 20$ points as our design points and another $10,000$ points as a candidate set. We then computed the logarithm of the income and created an indicator for each person: 1 if their log income is above 10 and 0 otherwise.

All of the variables education, age, sex, own child, health insurance, marital status, employment and disability are categorical and we hence used a one-hot encoding. The education variable is an ordinal variable but we encoded it as a continuous variable for convenience. We then sampled $n_{\text{seq}} = 200$ new points via sequential design using BART-Int and GP-BQ, and sampled randomly for MI.

As a baseline ground truth, we used all $454,816$ points in the dataset and estimated the mean via MI. We also double-checked by using BART-Int with $10,020$ points from the design and candidate sets, which yielded very similar results.

We repeated this set-up 20 times over different random initial points but the same candidate set. For GP-BQ, we set the lengthscale and $\sigma$ by maximising the marginal likelihood. For BART-Int, we mostly followed the default settings but used $T = 50$ trees, 1000 burn-in points, 5000 posterior draws after burn-in and kept every 3 draws from the posterior (thinning) so that $m = 1666$.