[Reviews · NeurIPS 2020]

Review 1

Summary and Contributions: This paper tackles the Bayesian quadrature problem adapting the BART as a prior. The new approach is compared to the standard method, which uses a Gaussian process prior.

Strengths: The motivation and problem setting were well defined and workings were clearly written. The novelty will be adapting the BART to numerical integration problems while the BART has been used to fit regressions. The convergence results of BPNI estimators using GP and BART priors are stated and simulation studies are provided. The additive-tree-structure prior has some strength on well-known limits (continuous predictor function and curse of dimension) and a computational burden of the GP prior

Weaknesses: Both GP and BART priors have been well known and, this paper applied those methods to a numerical integration problem. Base on my limited knowledge, this is the new study. The question is how much demand for solving numerical integration (in the Bayesian framework) is in the community. I think some case studies or problems that the NeurlPS community is familiar with, will make it easier to engage with the community. For the BPNI estimator using a GP prior, I couldn’t find how points (x’s) are generated.

Correctness: Although I didn’t closely looked at the mathematical proofs, I didn’t find any fault claim in the main draft.

Clarity: Overall, it was easy to follow and well structured. I found wordings “posterior distribution on \Pi[f]” and “posterior distribution on f” confusing. A clear mathematical notion would be helpful.

Relation to Prior Work: Authors included literature for GP and BART priors.

Reproducibility: No

Additional Feedback:


Review 2

Summary and Contributions: This manuscript proposes a new Bayesian numerical model, based on Bayesian Additive Regression Trees (BART) priors. The advantage of this new methodology is that all the knowledge about the problem can be straightforwardly implemented in a prior, and the posterior distribution can allow quantifying the uncertainty about the exact value of this quantity. The advantages and disadvantages of this new methodology are also solidly provided with experiments.

Strengths: It provides reasonable approach to improve the accuracy of numerical approximations, which could be potentially be utilized in the target field of study of application. The related work is also adequately referred. Limitation of the current GP-BQ approaches have been declared.

Weaknesses: However, the novelty of this approach is not solidly emphasized compared with previous works (Bayesian additive regression trees (BART) in the Bayesian probabilistic numerical integration field. The impact significance of is work is also vague and not clearly emphasized to the readers. Hence a major revision is suggested before being considered to be published in NIPS. For example, From the calculation algorithm, how does this method differ from BART method with similar application?

Correctness: yes

Clarity: The theoretical work in the paper needs to be written more clearly.

Relation to Prior Work: The novelty of this approach is not solidly emphasized compared with previous works (Bayesian additive regression trees (BART) in the Bayesian probabilistic numerical integration field.

Reproducibility: Yes

Additional Feedback: NA


Review 3

Summary and Contributions: "Bayesian Probabilistic Numerical Integration with Tree-Based Models" proposes using Bayesian Adaptive Random Trees (BART) in place of Gaussian processes for Bayesian quadrature.

Strengths: The paper reads well, and provides a pleasant introduction to the topic for the reader. The writing is clear and precise, and the contributions are easily found. The paper tackles an interesting and relevant problem for the ML community. The experiments are intuitive and relatively straightforward to understand. Theoretical guarantees are a welcome addition, although only shown asymptotically.

Weaknesses: I believe the main weakness of this work is a lack of real-world experiments compelling to the NeurIPS community. A secondary concern is the strength of the theoretical results presented. 1. Experiments The paper starts out with a discussion of all the use cases for numerical integration in the ML community in lines 15-19. This is great. Unfortunately the actual experiments presented do not draw from any of these application areas. The experimental results are mostly based on the Genz test functions, which are a set of toy problems first published all the way back in 1984. This seems like a good start, but not at all a convincing experimental result. The addition of step functions seems suspicious as well. I understand that BART-Int claims to offer superior performance on discontinuous functions, but then why not present results both with and without the added step functions? The "Bayesian Survey Design" experiment seems more promising as it is based on real data, but it is based on a dataset I'm not familiar with. What about running testing BART-Int on any of the concrete problems presented in the introduction? Also the paper appears to be missing any ablation on the parameters $m$. the number of MCMC steps, and $T$, the number of trees, which would appear significantly impactful variables both in terms of the performance as well as computational cost, O(Tmn). 2. Theory Theorem 1 is quite a mouthful, but IIUC it's providing only a relatively weak form of convergence. In particular the punchline seems to be asymptotic almost surely convergence. Shouldn't this be relatively obvious? As long as my incoming samples are drawn from \pi, my BART estimator should converge to the true function asymptotically. And if my estimate of the function is converging in the limit, then my estimate of its integral ought to be converging as well. Perhaps I'm missing something here, but this asymptotic guarantee doesn't tell me much beyond the fact that this is a consistent estimator. On a lesser note, I would like to add that more exposition around Theorem 1 would go a long way towards readability. The current discussion of the theorem focuses more on its applicability to prior work, which is great but not helpful for building intuition. In particular the result that P(|a - b| / C_n > eps_n) -> 0 for any C_n that goes to infinity seems like a weak result at first. I'm still not sure that I have fully grokked the full implications of the theorem.

Correctness: I have not verified the proofs personally. The empirical methodology seems correct.

Clarity: * In the abstract "BQ approach is inherently limited to cases where GP approximations can be done in an efficient manner, thus often prohibiting high-dimensional or non-smooth target functions." Isn't the scaling concern with GPs (and other kernel methods) generally the size of the dataset, not the dimensionality of the data itself? I see that this is clarified somewhat on lines 63-66, but I feel that harping on points 1 and 2 instead of 3 in the abstract would be more compelling for the reader. * On line 43, "However, we will see that not all Bayesian estimators have this property." it's not immediately obvious to the reader what "this property" refers to. * The choice of a, b notation in lines 89-90 is highly unusual. Why no \mu, \sigma^2? Also it appears a \mathcal is missing on line 90? * What's up with this magic 8 showing up in the equation on line 125? Also is this (1/8)T or 1/(8T)? * The distinction between "BPNI" and Bayesian quadrature is not at all clear to me. This seems like flag-planting more than useful terminology. * "for for" on line 237. * Some aspects of Table 1 are not clear. Does each line of the Setup column actually correspond with the rows in the rest of the table? The term "MI" is not introduced until further down in the text; a description in the caption would be ideal. Also "MC" seems like the more orthodox acronym. * At first glance it's not obvious what ground truth is in Figure 1. Adjusting the label or clarifying in the caption would be nice for the impatient reader.

Relation to Prior Work: Yes, prior work appears to be well addressed. Nothing blatantly missing as far as I'm aware.

Reproducibility: Yes

Additional Feedback:


Review 4

Summary and Contributions: The authors designs a Bayesian quadrature procedure for integral approximations using a BART regressor instead of a Gaussian Process (GP) regressor.

Strengths: The idea is nice with theoretical support. The topic is relevant for NeurIPS community.

Weaknesses: Although the idea is nice the degree of novelty is low (or very low).

Correctness: The derivations seem to be correct.

Clarity: Some important points must be clarified. The material is very compressed.

Relation to Prior Work: The study of related works is extremely poor. This is the main drawback of the paper. The authors must clarify the relationship with this work F. Llorente et al, "Adaptive quadrature schemes for Bayesian inference via active learning", arXiv:2006.00535, 2020 which contains very related material (the Nearest Neighbor part is strongly related). Other works should be also mentioned since consider piecewise constant surrogate functions to approximate integrals (using the piecewise constant surrogate functions in different way: even as proposal densities in Monte Carlo schemes), for instance: H. Ying, K. Mao, and K. Mosegaard, “Moving Target Monte Carlo,” arXiv preprint arXiv:2003.04873, 2020 T. E. Hanson, J. V. D. Monteiro, and A. Jara, “The Polya tree sampler: Toward efficient and automatic independent Metropolis–Hastings proposals,” Journal of Computational and Graphical Statistics, vol. 20, no. 1, pp. 41–62, 2011. L. Martino, R. Casarin, F. Leisen, and D. Luengo, “Adaptive independent sticky MCMC algorithms,” EURASIP Journal on Advances in Signal Processing, vol. 2018, no. 1, p. 5, 2018 J. Felip, N. Ahuja, and O. Tickoo, “Tree pyramidal adaptive importance sampling,” arXiv preprint arXiv:1912.08434, 2019.

Reproducibility: No

Additional Feedback: I like the idea proposed in the paper. My main concerns are two (which are also related): (a) the degree of novelty (b) and the poor state-of-the-art discussion. Mainly, the authors must clarify the relationship with this work F. Llorente et al, "Adaptive quadrature schemes for Bayesian inference via active learning", arXiv:2006.00535, 2020 which contains very related material (the Nearest Neighbor part is strongly related). Other works should be also mentioned since consider piecewise constant surrogate functions to approximate integrals (using the piecewise constant surrogate functions in different way: even as proposal densities in Monte Carlo schemes), for instance: H. Ying, K. Mao, and K. Mosegaard, “Moving Target Monte Carlo,” arXiv preprint arXiv:2003.04873, 2020 T. E. Hanson, J. V. D. Monteiro, and A. Jara, “The Polya tree sampler: Toward efficient and automatic independent Metropolis–Hastings proposals,” Journal of Computational and Graphical Statistics, vol. 20, no. 1, pp. 41–62, 2011. L. Martino, R. Casarin, F. Leisen, and D. Luengo, “Adaptive independent sticky MCMC algorithms,” EURASIP Journal on Advances in Signal Processing, vol. 2018, no. 1, p. 5, 2018 J. Felip, N. Ahuja, and O. Tickoo, “Tree pyramidal adaptive importance sampling,” arXiv preprint arXiv:1912.08434, 2019. Another issue of the paper is the clarity. See below some suggestions for improving the paper: - In the posterior mean g^n(x) at page 4, please clarify where the outputs y appear. - At page 5, we introduce a nice alternative solution which is quite related to the NN solution of the paper F. Llorente et al, "Adaptive quadrature schemes for Bayesian inference via active learning", arXiv:2006.00535, 2020. Please clarify this connection.


Review 5

Summary and Contributions: This paper proposes the use of BART for BPNI, which has certain computational advantages over the more-standard Gaussian processes. It also provides more general theoretical results about error bounds for BPNI which should be of broader interest.

Strengths: The use of BART in the context is, to my knowledge, novel. The authors provide pretty compelling evidence that BART-Int improves estimation error in BPNI over GP-BQ, particularly in higher dimensions and with integrands that are discontinuous. The empirical evidence is consistent with what I would expect, having used GPs and BART in a wide range of settings, and the authors do a good job explaining why BART-Int improves on BP-GP (in the settings where it does). Theorem 1 is also novel, and while not earth-shattering should be of interest to practitioners of BPNI. (I haven't been through the proof line by line but the statement is eminently plausible). I think the paper is quite relevant to the NeurIPS community.

Weaknesses: 1. At the outset of section 5, the authors state that "We emphasize that our goal is to compare methods which can provide a Bayesian quantification of uncertainty; as a result, it is very much possible that non-Bayesian methods could have better point estimate performance on some of these problems." But the performance metrics given are primarily about estimation error, not the calibration of Bayesian posteriors over the intergal or coverage of Bayesian intervals. So while BART-Int has good performance in estimation, particularly when the dimension is high and/or the integrand is discontinuous, I'm left wondering if it does provide better measures of uncertainty in general. Fig 1 and 3 do show some intervals, but if I understand them correctly they're for one simulation run. The empirical evidence in favor of BART-Int would be stronger if calibration/coverage took front stage, although the improvements in estimation error appear significant and notable. 2. The paper could talk more about the potential limitations of BART as a prior over f in this context. In particular, a Gaussian process with a standard covariance kernel (I'll assume squared exponential here) is going to interpolate and extrapolate in a well-understood fashion, with intervals over function evaluations that gradually increase as we leave the support of the data. A BART prior over f has a covariance function that decays much faster than the squared exponential, so the estimated function will tend to be much less smooth (obvious on plotting a BART fit!). Standard BART priors are going to interpolate and extrapolate using step functions, which may or may not work well depending on the integrand. Taken together this means that the intervals around an evaluation of f far from the support of data will tend to be narrow relative to a Gaussian process, although they will often be wider extrapolating *near* the support of the data because of the lack of smoothness. And what's relevant here is a sum over function evaluations at many locations xtilde -- uncertainty in that sum will depend on the marginal variances over function evaluations as well as the correlation between the function evaluations, and it's not completely clear how they will net out -- to wit, the GP tends to give narrower intervals in the survey design application and wider intervals in fig 1. All that said, It's clear from their experiments that using BART in this context is doing something reasonable and improving in a meaningful way on existing methods, at least on some metrics and in some (relevant) contexts. I don't think either of these weaknesses are fatal or should preclude acceptance.

Correctness: Yes (see comments above about Thm 1)

Clarity: Yes.

Relation to Prior Work: Yes. The coverage of the general Bayesian tree literature is good for a paper of this scope. I'm not expert in BPNI but the coverage of that literature also seems good.

Reproducibility: Yes

Additional Feedback: The paper notes that BART is "similar in spirit and effectiveness to random forests". A minor point, but it's much closer in spirit to boosting than to random forests.

[Author Response · NeurIPS 2020]

We would like to thank reviewers for their time and effort in providing us with feedback. Please find our response below, which focuses on the major points discussed by reviewers (R1, R2, R3 & R4).

**Contribution/Relevance to the community (R1 & R4):** Reviewer 1 asks "how much demand for solving numerical integration (in the Bayesian framework) is in the community". We would argue the demand is significant! See [1,3,21,26,28,35,47,58,73] which were all published at leading machine learning conferences, and [4,11,33,36,37,38,49,50] which appeared at leading venues in computational statistics or applied mathematics. We propose to further clarify this point, and add additional references in the machine learning literature.

**Related Literature (R4):** We thank R4 for the opportunity to expand. The novelty of our paper is to use tree-based models which are inherently Bayesian, and could hence be used for quantifying integration error in a Bayesian manner (as per the BPNI framework). This is different from the suggested references, where trees are used for MCMC proposals, which is not Bayesian per se. Furthermore, those papers use trees to approximate a density rather than the integrand. However, we agree that this literature is relevant and could motivate further research in BPNI. We propose to expand significantly on similarities and differences, and thank R4 for challenging us on this point.

The criticism about the Llorente et al. paper is quite unfair given that this paper appeared online *after* the NeurIPS abstract submission deadline. That said, we will discuss the nearest-neighbours approach, which could also fall within the BPNI framework (except that this paper only considers point estimates, rather than entire distribution). From the point of view of the models, the main difference is the way in which splits are performed. BART will adapt to the smoothness and sparsity of $f$ in a way that the nearest neighbours approach cannot. We also note that the rate of convergence presented in that paper is much slower than for BART-Int.

**Theory (R3):** The result is written in a general form so that it can be used to provide stronger results than consistency: it allows for rates of convergence. This rate will depend on (i) the point set, and (ii) the prior model. The form of this theorem allows us to understand the specific impact of these two aspects, and as a result understand how the method will perform relative to competitors, and potentially how to improve it. We propose to further discuss these points, and to unpack further some of the more complex mathematical details.

**Experiments (R1 & R3):** The experiments considered have all previously been used as benchmarks by the community. The Genz functions are particularly useful as they can highlight strengths/weaknesses of different methods. The survey design problem with a Bayesian lense first appeared in an ICML paper; see [23].

Of course, further experiments could be useful, but we were not able to do this due to space constraints. Since reviewers agree that it would improve the paper, we propose to include new examples in the supplementary material, focusing on modern ML benchmarks, e.g. estimating integrals and evidence for Bayesian inference and model selection (e.g. Chai et al. (2019) and Gunter et al. (2014)), and uncertainty quantification in applied settings (e.g. Oates et al. (NeurIPS 2017)).

**Other comments:** Thank you for the additional feedback; we will clarify these points. Specifically:

- **R1:** Thanks, we will clarify the notation for the posterior distributions on $f$ and $\Pi[f]$.
- **R3:** We used BPNI rather than BQ since our estimator is not a quadrature rule (this means a linear combination of function values). Our terminology was used in [11].
- **R3:** GPs also struggle with slow convergence rates in high d settings [70,72]. Some tree-based models can make use of sparsity structures to avoid this issue; see [41].

[Meta-Review · NeurIPS 2020]

Based on the level of disagreement in the initial reviews -- and due to concerns brought up about novelty of the approach -- I invited a fifth reviewer, with strong experience in the area, to review the paper. This review reinforced the novelty of the approach, and agreed with its usefulness. Overall, this is a paper that should be of interest to the Bayesian Optimization community. However, we strongly suggest that the authors take into account the reviewers' comments on clarity in preparing the final version. In addition to the points mentioned in the rebuttal, I would like to see discussion of the limitations of BART in this context, in particular regarding out-of-sample uncertainty estimates, as brought up by reviewer 5. While we do not expect the authors to have compared against concurrent work, we think it would improve the paper to discuss the related work mentioned by Reviewer 3.